# Geminiviruses encode additional small proteins with specific subcellular localizations and virulence function

Pan Gong [1,6], Huang Tan [2,3,6], Siwen Zhao[1], Hao Li[1], Hui Liu[4], Yu Ma[2,3], Xi Zhang[2,3], Junjie Rong[2,3], Xing Fu[2], Rosa Lozano-Durán [2,5✉], Fangfang Li[1✉] & Xueping Zhou [1,4✉]

Geminiviruses are plant viruses with limited coding capacity. Geminivirus-encoded proteins are traditionally identified by applying a 10-kDa arbitrary threshold; however, it is increasingly clear that small proteins play relevant roles in biological systems, which calls for the reconsideration of this criterion. Here, we show that geminiviral genomes contain additional ORFs. Using tomato yellow leaf curl virus, we demonstrate that some of these small ORFs are expressed during the infection, and that the encoded proteins display specific subcellular localizations. We prove that the largest of these additional ORFs, which we name V3, is required for full viral infection, and that the V3 protein localizes in the Golgi apparatus and functions as an RNA silencing suppressor. These results imply that the repertoire of geminiviral proteins can be expanded, and that getting a comprehensive overview of the molecular plant-geminivirus interactions will require the detailed study of small ORFs so far neglected.

[1] State Key Laboratory for Biology of Plant Diseases and Insect Pests, Institute of Plant Protection, Chinese Academy of Agricultural Sciences, Beijing, China. [2] Shanghai Center for Plant Stress Biology, CAS Center for Excellence in Molecular Plant Sciences, Chinese Academy of Sciences, Shanghai, China. [3] University of the Chinese Academy of Sciences, Beijing, China. [4] State Key Laboratory of Rice Biology, Institute of Biotechnology, Zhejiang University, Hangzhou, Zhejiang, China. [5] Department of Plant Biochemistry, Center for Plant Molecular Biology (ZMBP), Eberhard Karls University, Tübingen, Germany. [6] These authors contributed equally: Pan Gong, Huang Tan. ✉email: rosa.lozano-duran@zmbp.uni-tuebingen.de; lifangfang@caas.cn; zzhou@zju.edu.cn

Viruses are intracellular parasites that heavily rely on the host cell machinery to complete their infectious cycle. Most viruses have small genome sizes, with the concomitant limitation in coding capacity; in order to overcome the restrictions imposed by their reduced proteome, viruses have evolved to encode multifunctional proteins that efficiently target hub proteins in their host cells (reviewed in[1,2]). Nevertheless, higher numbers of virus-encoded proteins might enable more sophisticated infection mechanisms, and therefore maximization of the coding space would be expected to be an advantage to the pathogen.

Geminiviruses are a family of plant viruses with circular, single-stranded (ss) DNA genomes causing devastating diseases in crops around the globe. This family includes nine genera, based on host range, insect vector, and genome structure: *Becurtovirus*, *Begomovirus*, *Curtovirus*, *Eragrovirus*, *Mastrevirus*, *Topocuvirus*, *Turncurtovirus*, *Capulavirus*, and *Grablovirus*[3]; most species described to date belong to the genus *Begomovirus*. Members of this family have small genomes, composed of one or two DNA molecules of less than 3 Kb each, in which the use of coding space is optimized by bidirectional and partially overlapping open reading frames (ORFs): in one <3 Kb molecule, geminiviruses contain up to 7 ORFs, with a known maximum of 8 viral proteins per virus. The geminiviral infection cycle is complex, and multiple steps remain to be fully elucidated. Following transmission by an insect vector, the geminiviral DNA genome must be released from the virion and reach the nucleus, where it will be converted into a double-stranded (ds) DNA replicative intermediate; this dsDNA molecule will serve as template for the transcription of viral genes, including the replication-associated protein (Rep), which reprograms the cell cycle and recruits the host DNA replication machinery. Rolling-circle replication ensues, by which new ssDNA copies of the viral genome are produced. Eventually, the virus must move intracellularly, intercellularly, and systemically, invading new cells and making virions available for acquisition by the vector. In order to accomplish a successful infection, geminiviruses must tailor the cellular environment to favor their replication and spread; for this purpose, they modify the transcriptional landscape of the infected cell, re-direct post-transcriptional modifications, and interfere with hormone signaling, among other processes (reviewed in[4–6]), ultimately suppressing anti-viral defenses, creating conditions favorable to viral replication, and manipulating plant development. Although geminivirus-encoded proteins are described as multifunctional, how the plethora of tasks required for a fruitful infection can be performed by only 4–8 proteins is an intriguing biological puzzle. Whether members of this family encode additional small proteins, below the arbitrary 10 kDa threshold established following identification of the first geminivirus species, remains elusive.

Tomato yellow leaf curl virus (TYLCV) is a monopartite begomovirus and the causal agent of the destructive tomato leaf curl disease[7]. The TYLCV genome contains six known open reading frames (ORFs), encoding the capsid protein (CP)/V1 and V2 in the virion strand, and C1/Rep, C2, C3, and C4 in the complementary strand. Rep creates a cellular environment permissive for viral DNA replication and attracts the DNA replication machinery to the viral genome (reviewed in[8]); C2 suppresses post-transcriptional gene silencing (PTGS)[9], protein ubiquitination[10] and jasmonic acid (JA) signaling[10,11]; C3 interacts with PROLIFERATING CELL NUCLEAR ANTIGEN (PCNA), the NAC family transcription factor SlNAC1, and the regulatory subunits of DNA polymerases α and δ to enhance viral replication[12–14]; C4 is a symptom determinant, interferes with the intercellular movement of PTGS, and hampers salicylic acid (SA)-dependent defenses[9,15,16]; the CP forms the viral capsid and is essential for the transmission by the insect vector, and shuttles the viral DNA between the nucleus and the cytoplasm[17–24]; V2 is a strong suppressor of PTGS as well as transcriptional gene silencing (TGS), and it mediates the nuclear export of CP[25–29]. Interestingly, a recent report identified 21 transcription initiation sites within the TYLCV genome by taking advantage of cap-snatching by rice stripe virus (RSV) in the experimental *Solanaceae* host *Nicotiana benthamiana*, suggesting that transcripts beyond those encoding these known ORFs might exist[30]. This idea is further indirectly supported by the fact that attempts at knocking-in tags in geminiviral genomes have so far been fruitless.

Here, we report that geminiviral genomes contain additional ORFs besides the canonical ones described to date. These previously neglected ORFs frequently encode proteins that are phylogenetically conserved. Using the geminivirus TYLCV as an example, we show that some of these ORFs are transcribed during the viral infection, and that the proteins they encode accumulate in the plant cell and show specific subcellular localizations and distinctive features. Moreover, we demonstrate that one of these additional ORFs, which we have named V3 and is conserved in begomoviruses, is essential for full infectivity in *N. benthamiana* and tomato, and encodes a Golgi-localized protein that acts as a suppressor of PTGS and TGS. Taken together, our results indicate that geminiviruses encode additional proteins to the ones described to date, which may largely expand the geminiviral proteome and hence the intersection of these viruses with the host cell.

## Results

**The TYLCV genome contains additional conserved open reading frames**. It is increasingly clear that small proteins (<100 aa) are prevalent in eukaryotes, including plants, and have biological functions (reviewed in[31–33]). In order to explore whether geminiviral genomes may contain additional ORFs encoding small proteins of predictable functional relevance, we designed a tool that we called ViralORFfinder; this tool uses the ORFfinder from NCBI (https://www.ncbi.nlm.nih.gov/orffinder/) to identify ORFs in an inputted subset of DNA sequences (geminiviral genomes, in this case) and creates a small database with the translated protein sequences, which can be used to BLAST a protein of choice, therefore assessing conservation among the subset of selected species (Fig. 1a). The distribution of the protein of interest is then displayed in a phylogenetic tree of the inputted viral species, generated based on the DNA sequences provided. Using ViralORFfinder, additional ORFs can be consistently predicted in geminiviruses of different genera, as illustrated for bipartite begomoviruses (Supplementary Fig. 1; Supplementary Table 2), curtoviruses (Supplementary Fig. 2; Supplementary Table 3), and mastreviruses (Supplementary Fig. 3; Supplementary Table 4); the proteins encoded by some of these ORFs are conserved among species.

We then used ViralORFfinder to identify additional proteins encoded by monopartite begomoviruses causing tomato leaf curl disease isolated from different regions of the world (see "Methods" section; Supplementary Fig. 4; Supplementary Table 5), and identify those present in TYLCV and conserved in other species. As shown in Fig. 1b, 43 ORFs encoding proteins/peptides of >10 aa were identified in the TYLCV genome. Interestingly, a significant correlation can be found between the size of the encoded proteins and their representation in the selected subset of species, with the six larger proteins (>10 kDa) present in all of them (Supplementary Fig. 4b, pink). For further analyses, we selected 6 ORFs based on size and prevalence (Supplementary Fig. 4b, blue), named ORF1-6; the position of these ORFs in the

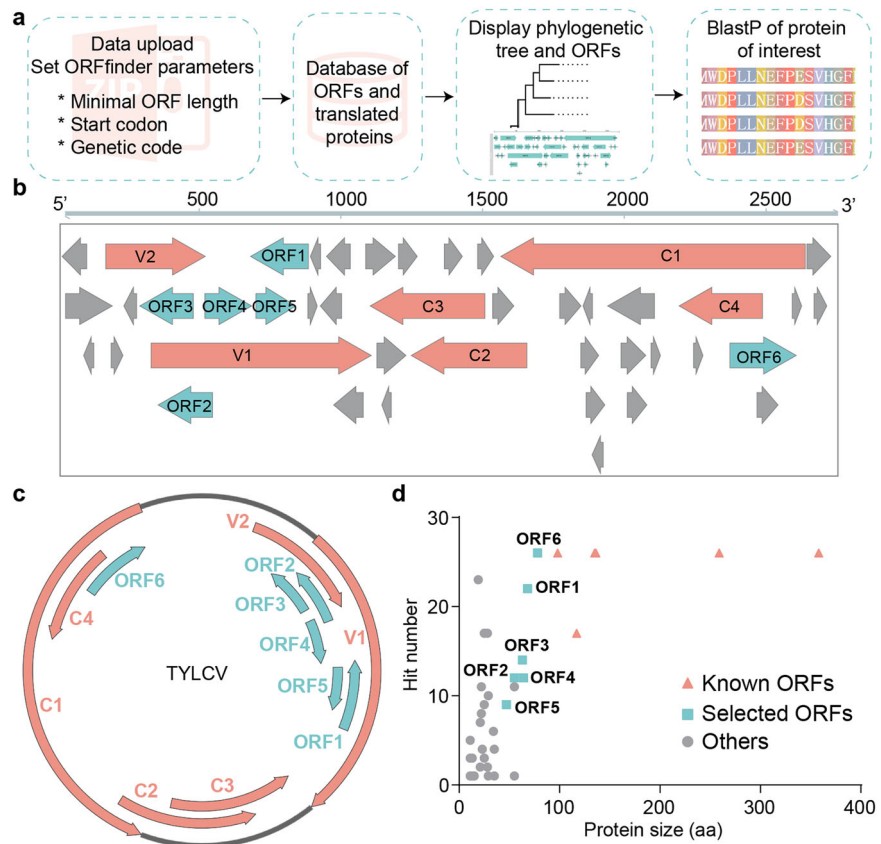

**Fig. 1 The TYLCV genome contains additional conserved open reading frames. a** Working pipeline of ViralORFfinder, a web-based tool for ORF prediction and protein conservation analysis. **b** Schematic view of predicted ORFs (≥ 30 nt) in the TYLCV genome. **c** Genome organization of TYLCV; arrows indicate ORFs. In (**b**) and (**c**), pink arrows represent the six known ORFs (C1, C2, C3, C4, V1, and V2), while blue arrows represent the six additional ORFs described in this work (ORF1-6). **d** Correlation between the size of proteins encoded by the ORFs in the TYLCV genome and their representation in the selected subset of begomoviruses (see Supplementary Table 6). The TYLCV isolate used in these experiments is TYLCV-Alm.

TYLCV genome is shown in Fig. 1c. Of note, the proteins encoded by these ORFs are also conserved in other members of the *Begomovirus* genus, both bipartite and monopartite, infecting a broad range of hosts (Supplementary Fig. 5–7; Supplementary Table 6; Fig. 1d).

**Additional ORFs in TYLCV encode proteins with predicted domains and specific subcellular localizations.** With the aim of gaining insight into the properties of the proteins encoded by the additional ORFs from TYLCV, we investigated the presence in their sequence of predicted domains or signals, namely transmembrane domains (TM), nuclear localization signals (NSL), and chloroplast transit peptides (cTP). As shown in Fig. 2a, while none of these proteins contains an NLS, and only one of them contains a cTP, three of them contain a predicted TM, which is not present in any of the previously characterized proteins.

We then cloned ORF1-6, fused them to the GFP gene, and transiently expressed them in *N. benthamiana* leaves. Confocal microscopy indicates that these fusion proteins present specific subcellular localizations (Fig. 2b). Whereas the ORF1-encoded protein is mostly nuclear, co-expression with marker proteins or dyes unveils that the ORF2-encoded protein localizes in the endoplasmic reticulum (ER), as demonstrated by the co-localization with RFP-HDEL; the ORF3-encoded protein in mitochondria, as demonstrated by the co-localization with MitoTracker; the ORF4-encoded protein in the ER and the Golgi apparatus, as demonstrated by the partial co-localization with RFP-HDEL and SYP32-RFP; and the ORF5- and ORF6-encoded

proteins mostly in Golgi, as demonstrated by the partial co-localization with SYP32-RFP (Fig. 2c–g; Supplementary Fig. 8). The specific subcellular localization exhibited by each of these proteins suggests that, despite their small size (5.3-9.3 kDa; Fig. 2a), either their sequence contains the appropriate targeting signals, or they interact with plant proteins that enable their precise targeting in the cell.

A prerequisite for these ORFs to have a biological function is their expression in the context of the viral infection. Therefore, we cloned the ~500-bp TYLCV genomic sequence upstream of each ATG (pORF1-6) before the GFP reporter gene and tested its promoter activity in transiently transformed *N. benthamiana* leaves. As shown in Supplementary Fig. 9, none of these sequences could drive GFP expression in the absence of the virus, but pORF1, pORF2, pORF4, and pORF5 could when the virus was present; the sequence upstream of the C4 ORF was used as positive control, and could activate GFP expression both in the presence and absence of TYLCV. The promoter activity of these upstream sequences in infected cells strongly suggests that at least ORF1, 2, 4, and 5 are expressed during the viral infection.

**The V3 protein from TYLCV is a Golgi-localized silencing suppressor required for full infection.** ORF6 is the largest of the newly described ORFs in the TYLCV genome, and the protein it encodes displays the highest degree of conservation in a selected subset of 26 representative begomoviruses (Supplementary Fig. 7). Therefore, we decided to further characterize this ORF as a proof-of-concept of the potential biological roles of additional

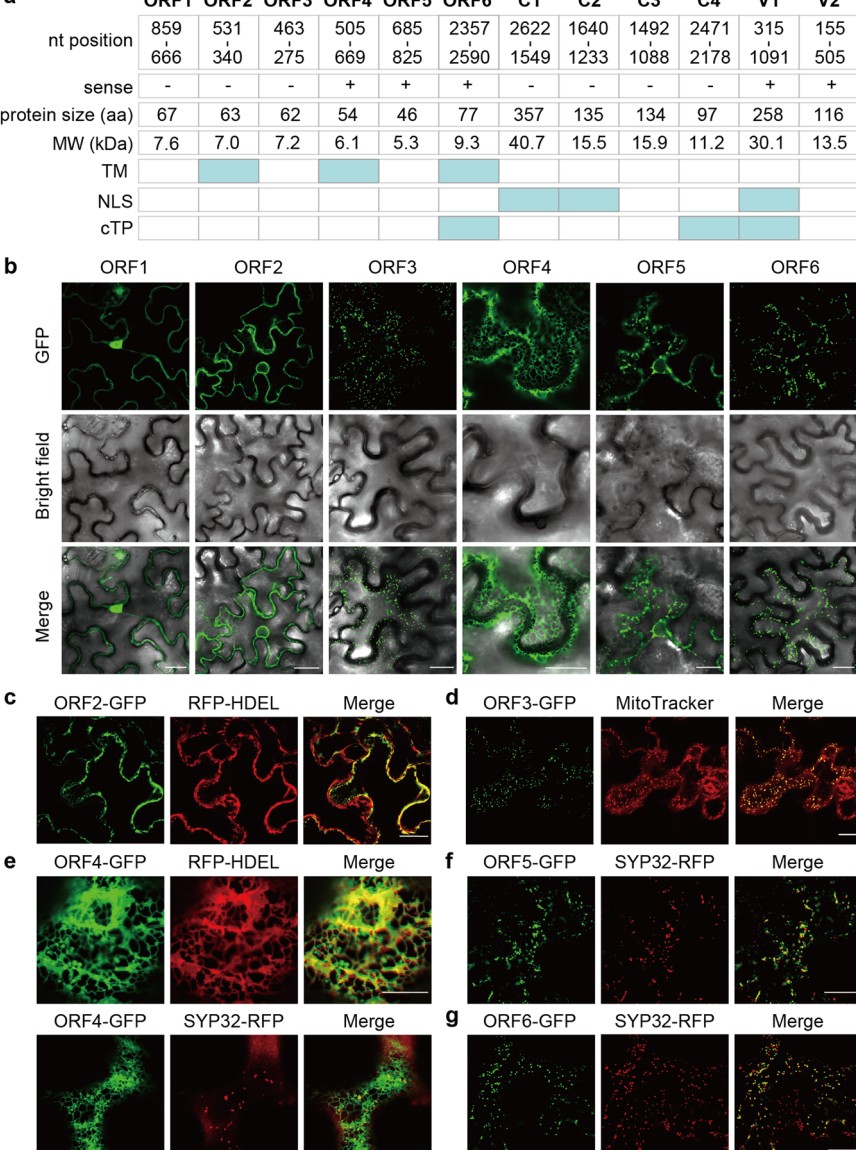

| | ORF1 | ORF2 | ORF3 | ORF4 | ORF5 | ORF6 | C1 | C2 | C3 | C4 | V1 | V2 |
|---|---|---|---|---|---|---|---|---|---|---|---|---|
| nt position | 859 - 666 | 531 - 340 | 463 - 275 | 505 - 669 | 685 - 825 | 2357 - 2590 | 2622 - 1549 | 1640 - 1233 | 1492 - 1088 | 2471 - 2178 | 315 - 1091 | 155 - 505 |
| sense | - | - | - | + | + | + | - | - | - | - | + | + |
| protein size (aa) | 67 | 63 | 62 | 54 | 46 | 77 | 357 | 135 | 134 | 97 | 258 | 116 |
| MW (kDa) | 7.6 | 7.0 | 7.2 | 6.1 | 5.3 | 9.3 | 40.7 | 15.5 | 15.9 | 11.2 | 30.1 | 13.5 |
| TM | | ▇ | | ▇ | | ▇ | | | | | | |
| NLS | | | | | | | ▇ | ▇ | | | ▇ | |
| cTP | | | | | | ▇ | | | | ▇ | ▇ | |

**Fig. 2 Additional open reading frames in the TYLCV genome encode proteins with predicted domains and specific subcellular localizations. a** Nucleotide position of the six additional ORFs and the six known ORFs in the TYLCV genome, size (in aa) and predicted molecular weight (MW; in kDa) of the corresponding encoded proteins, and domains or signals predicted in the protein sequence. TM: transmembrane domain; NLS: nuclear localization signal; cTP: chloroplast transit peptide. **b** Subcellular localization of the proteins encoded by ORF1-6 fused to GFP at their C-terminus transiently expressed in *N. benthamiana* leaves. Scale bar: 25 µm. **c** Co-localization of ORF2-GFP with the ER marker RFP-HDEL. Scale bar: 25 µm. **d** Co-localization of ORF3-GFP with the mitochondrial stain MitoTracker Red. Scale bar: 25 µm. **e** Co-localization of ORF4-GFP with the ER marker RFP-HDEL and the cis-Golgi marker SYP32-RFP. Scale bar: 10 µm. **f** Co-localization of ORF5-GFP with the cis-Golgi marker SYP32-RFP. Scale bar: 25 µm. **g** Co-localization of ORF6-GFP with the cis-Golgi marker SYP32-RFP. Scale bar: 25 µm. These experiments were repeated at least three times with similar results; representative images are shown. The TYLCV isolate used in these experiments is TYLCV-Alm.

ORFs. Hereafter, ORF6 will be referred to as *V3*, since it is the third ORF on the viral strand in the TYLCV genome.

The *V3* ORF is located in positions 2350–2583 of the TYLCV genome (TYLCV-BJ; Fig. 1d), and encodes a 77-amino acid protein. In different begomovirus species, the *V3* ORF ranges from 87 to 234 nt, and the protein it encodes presents a high degree of similarity, with 6 residues completely conserved (Supplementary Fig. 7).

In order to determine whether *V3* is transcribed during the viral infection, we checked if the corresponding transcript was present in TYLCV-infected samples: as shown in Fig. 3a, the *V3* transcript was found upon TYLCV infection, but not in uninfected plants. 5′ rapid amplification of cDNA ends (RACE)

was then used to identify the transcriptional initiation site of *V3*, which was found to be located between 176 and 421 nt upstream of the start codon (Fig. 3b). Interestingly, most of the sites identified by RACE (7 out of 11) are close to A2058, which was previously isolated by cap-snatching of RSV[30].

The finding that a transcript corresponding to the *V3* ORF can be identified in infected samples strongly suggests that the sequence upstream of this ORF must act as a promoter. Since the 500 bp fragment previously tested (Supplementary Fig. 9) did not show promoter activity, we tested a larger, 833-nt sequence upstream of the *V3* start codon, which was cloned before the GUS or GFP reporter genes. As shown in Fig. 3c, d, this viral sequence could activate GUS expression, leading to detectable GUS activity,

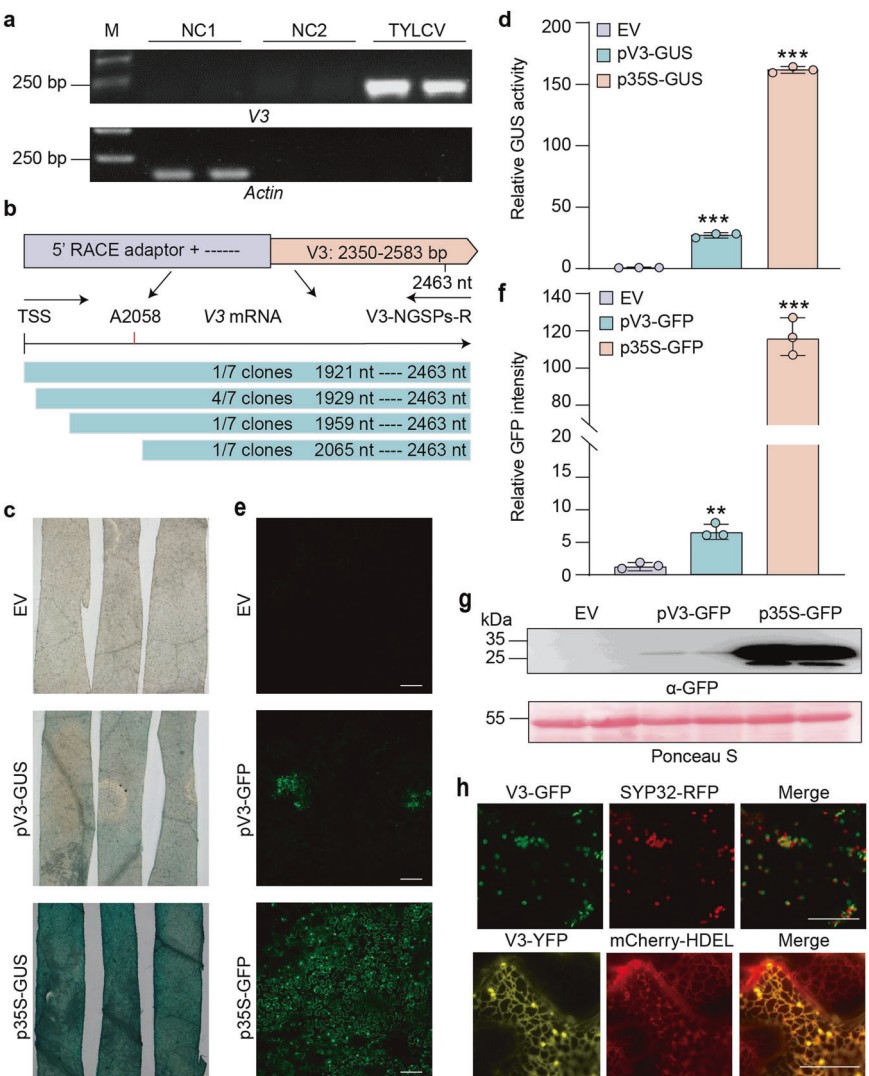

**Fig. 3 ORF6/V3 is expressed during the infection and encodes a Golgi-localized protein. a** RT-PCR analysis of *V3* transcripts from TYLCV-infected or uninfected *N. benthamiana* plants. M: DNA ladder marker. NC1: negative control 1 (reverse-transcription of total RNA extracted from uninfected plants with RT Primers). NC2: negative control 2 (reverse-transcription of total RNA extracted from uninfected plants with *V3*-specific primers). TYLCV: reverse-transcription of total RNA extracted from TYLCV-infected plants with *V3*-specific primers. This experiment was repeated three times with similar results; representative images are shown. **b** Transcriptional start site analysis of TYLCV V3 by 5′ RACE. TSS: transcription start site. A2058: V3 TSS captured by RSV cap-snatching (Lin et al., 2017). **c** Activity of pV3 promoter (and p35S promoter as positive control) in promoter-GUS fusions in transiently transformed *N. benthamiana* leaves at 2 dpi. EV: empty vector. This experiment was repeated three times with similar results; representative images are shown. **d** Quantification of relative GUS activity in samples from (**c**). Data are the mean of three independent biological replicates. Error bars represent SD. An asterisks indicate a statistically significant difference according to unpaired Student's *t* test (two-tailed), *** $p < 0.001$. **e** Activity of pV3 promoter (and p35S promoter as positive control) in promoter-GFP fusions in transiently transformed *N. benthamiana* leaves at 2 dpi. Scale bar: 100 μm. EV: empty vector. This experiment was repeated three times with similar results; representative images are shown. **f** Quantification of relative GFP intensity in samples from (**e**). Data are the mean of three independent biological replicates. Error bars represent SD. Asterisks indicate a statistically significant difference according to unpaired Student's *t* test (two-tailed), ** $p < 0.01$, *** $p < 0.001$. **g** Western blot analysis of GFP protein from (**e**). Ponceau S staining of the large RuBisCO subunit serves as loading control. EV: empty vector. This experiment was repeated three times with similar results; representative images are shown. **h** Co-localization of V3-GFP with the cis-Golgi maker SYP32-RFP (upper panel) and co-localization of V3-YFP with the ER maker mCherry-HDEL (lower panel). Scale bar: 20 μm. The TYLCV isolate used in these experiments is TYLCV-BJ. The original data from all experiments and replicates can be found in the Source data file.

and it could also drive expression of GFP (Fig. 3e–g), albeit more weakly than the 35S promoter. Taken together, these results confirm that the *V3* ORF can be expressed in planta from its genomic context.

Given that protein function is tightly linked to spatial location, we then decided to analyse the subcellular localization of the V3 protein in detail. As can be seen in Fig. 3h, V3 is a Golgi-localized protein; this localization does not change in the presence of the

virus (Supplementary Fig. 10a). A closer observation confirms that V3 is localized to cis-Golgi, as shown by the co-localization of V3-GFP with the markers Man49-mCherry and SYP32-RFP (Fig. 3h; Supplementary Fig. 10b); nevertheless, the YFP fusions YFP-V3 and V3-YFP can also partially co-localize with the endoplasmic reticulum (ER), as indicated by their partial co-localization with the ER marker mCherry-HDEL or RFP-HDEL (Fig. 3h; Supplementary Fig. 10c, d), which may

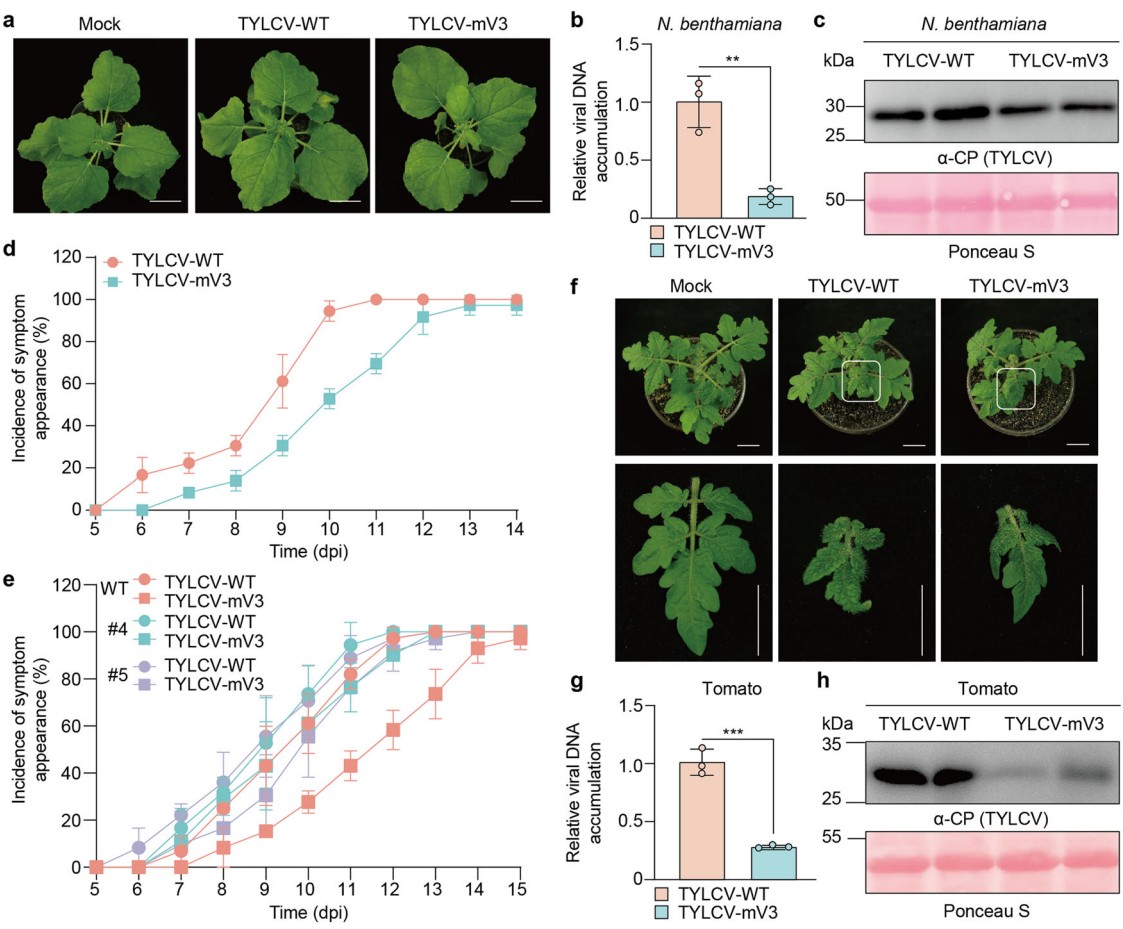

**Fig. 4 V3 is required for full TYLCV infection in *N. benthamiana* and tomato. a** Symptoms of *N. benthamiana* plants inoculated with wild-type TYLCV (TYLCV-WT), a V3 null mutant TYLCV (TYLCV-mV3), or mock-inoculated (pCAMBIA2300 empty vector), at 10 dpi. Bar = 2 cm. **b** Viral DNA accumulation in TYLCV-WT- and TYLCV-mV3-infected plants in (**a**), measured by qPCR. Data are the mean of three independent biological replicates. Error bars represent SD. An asterisk indicates a statistically significant difference according to unpaired Student's *t* test (two-tailed), *** $p < 0.01$. 25S RNA was used as internal reference. **c** Western blot showing TYLCV CP accumulation in systemic leaves of TYLCV-WT- and TYLCV-mV3-infected plants from (**a**). Ponceau S staining of the large RuBisCO subunit serves as loading control. This experiment was repeated three times with similar results; representative images are shown. **d, e** Incidence of symptom appearance in WT (**d, e**) or V3 transgenic (**e**) *N. benthamiana* plants infected with TYLCV-WT or TYLCV-mV3. Data are the mean of three independent biological experiments. Error bars represent SD. At least 8 plants were used per viral genotype and experiment. For images of symptoms in the V3 transgenic lines, see Supplementary Fig. 10d. **f** Symptoms of tomato plants inoculated with WT TYLCV (TYLCV-WT), a V3 null mutant TYLCV (TYLCV-mV3), or mock-inoculated (pCAMBIA2300 empty vector), at 10 dpi. Bar = 2 cm. **g** Viral DNA accumulation in TYLCV-WT- and TYLCV-mV3-infected plants from (**f**), measured by qPCR. Data are the mean of three independent biological replicates. Error bars represent SD. An asterisk indicates a statistically significant difference according to unpaired Student's *t* test (two-tailed), *** $p < 0.001$. 25S RNA was used as internal reference. **h** Western blot showing TYLCV CP accumulation in systemic leaves of TYLCV-WT- and TYLCV-mV3-infected plants from (**f**). Ponceau S staining of the large RuBisCO subunit serves as loading control. This experiment was repeated three times with similar results; representative images are shown. The TYLCV isolate used in these experiments is TYLCV-BJ. The original data from all experiments and replicates can be found in the Source data file.

reflect the transition of the protein from the ER to the Golgi apparatus.

With the aim of assessing the biological relevance of the V3 protein for the TYLCV infection, we next generated a mutated infectious clone carrying a T2351C substitution in the *V3* ATG, hence impairing the production of the V3 protein. Since the *V3* ORF overlaps with the Rep/C1 and C4 ORFs, nt replacements in the start codon of the *V3* ORF necessarily affect the protein sequence of the resulting Rep or C4 proteins; the chosen change results in a I89V substitution in the Rep/C1 protein, with no change in C4. This mutant infectious clone is hereafter referred to as TYLCV-mV3.

TYLCV-mV3 was then inoculated into *N. benthamiana* plants, and its performance compared to that of the wild-type (WT) virus (TYLCV-WT). At 10 days post-inoculation (dpi), TYLCV-

mV3-infected plants displayed mild leaf curling symptoms and presented lower viral DNA load compared to plants inoculated with TYLCV-WT (Fig. 4a, b), which correlated with a lower accumulation of CP (Fig. 4c). These differences apparently result from a delay in the infection, measured as symptom appearance (Fig. 4d). To evaluate the potential functional impact of the I89V substitution in Rep/C1 on the viral infection and disentangle this effect to that derived from the absence of V3, transgenic *N. benthamiana* lines expressing V3-YFP under a 35S promoter were generated, and a complementation assay was performed. Of note, the V3-expressing plants do not display obvious developmental abnormalities (Supplementary Fig. 11a); the expression of V3-YFP was confirmed by qRT-PCR and western blot (Supplementary Fig. 11b, c). As shown in Fig. 4e and Supplementary Fig. 11d, transgenic expression of V3-YFP could fully

complement the lack of V3 in the TYLCV-mV3 clone, measured as incidence and severity of symptom appearance, indicating that the slower progression of the infection observed upon inoculation with the V3 null mutant is due to the lack of this protein, and not to a suboptimal performance of Rep-I89V. Next, we evaluated the virulence of TYLCV-mV3 on tomato, the virus' natural host. As previously observed in *N. benthamiana*, the lack of V3 resulted in lower viral load and CP accumulation, and milder symptoms at 10 dpi (Fig. 4f–h), confirming that V3 plays a relevant role in the viral infection that is not restricted to *N. benthamiana*.

Heterologous expression from a potato virus X (PVX)-derived vector and quantification of the impact on PVX pathogenicity is a widely used approach to test virulence activity of viral genes of interest. Confirming a contribution of V3 to virulence, the presence of this gene in PVX-V3 led to an exacerbation of disease symptoms in inoculated *N. benthamiana* plants compared to PVX alone at 10 and 30 dpi, with a concomitant higher accumulation of the PVX CP at 30 dpi (Supplementary Fig. 12); βC1, a symptom determinant encoded by tomato yellow leaf curl China betasatellite (TYLCCNB), was used as positive control. Infection by PVX-V3, however, did not lead to $H_2O_2$ accumulation or cell death (Supplementary Fig. 12c).

It has been previously established that a high correlation exists between the ability of a viral protein to suppress RNA silencing and its capacity to enhance the severity of the PVX infection. RNA silencing is conserved in eukaryotes, and is considered the main anti-viral defense mechanism in plants[34–36]. Supporting this notion, virtually all plant viruses described to date encode at least one protein with RNA silencing suppression activity[36,37]. With the aim to test if V3 can suppress PTGS, we transiently expressed GFP from a 35S promoter in leaves of transgenic 16c *N. benthamiana* plants, harboring a 35S:GFP cassette[38], in the presence or absence of V3; the well-described silencing suppressor P19 from tomato bushy stunt virus (TBSV) was used as positive control. At 4 dpi, fluorescence had already substantially decreased when no viral protein was co-expressed, but was maintained in the samples with P19 or Myc-V3 (Fig. 5a). Western blot and qRT-PCR were used to confirm that both the GFP protein as well as the corresponding mRNA accumulated to higher levels in tissues expressing P19 or Myc-V3 (Fig. 5b, c). At 20 dpi, systemic leaves of 16c plants inoculated with the constructs to express either of the viral proteins remained green, while fluorescence had disappeared in control plants as a result of systemic silencing (Fig. 5a).

The ability of V3 to suppress PTGS was further confirmed by expressing this protein from a PVX-based vector; in this case, βC1, which also functions as silencing suppressor, was used as positive control. Transient co-transformation of a PVX infectious clone together with a 35S:GFP cassette in leaves of 16c plants led to weak fluorescence in the infiltrated tissues at 7 dpi, and systemic silencing at 20 dpi; in stark contrast, co-transformation with PVX-βC1 or PVX-V3 resulted in the maintenance of strong fluorescent signal at 7 dpi, and absence of systemic silencing (Fig. 5d). Neither βC1 nor V3 enhanced the local accumulation of PVX, as indicated by the accumulation of the PVX CP protein (Fig. 5e), hence ruling out an indirect effect of these proteins on the endogenous ability of PVX to suppress silencing. Therefore, our results demonstrate that V3 from TYLCV can effectively suppress PTGS in plants.

Another level of RNA silencing is TGS, which acts through methylation of DNA at cytosine residues; TGS also acts as an anti-viral response in plants, and is particularly relevant against geminiviruses, which replicate their DNA genomes in the nucleus of the infected cell[36,39]. To investigate whether V3 can also act as a TGS suppressor, we inoculated 16-TGS plants, in which the GFP transgene is silenced due to methylation of the 35S

promoter[40], with PVX or PVX-V3; PVX-βC1 was used as positive control, since βC1 can also act as a TGS suppressor. At 10 dpi, green fluorescence could be observed in systemic leaves of 16-TGS plants inoculated with PVX-βC1 and PVX-V3, as opposed to mock- or PVX-inoculated plants (Fig. 5f); this fluorescence persisted at 28 dpi (Fig. 5g). Visual assessment was confirmed at the molecular level by western blot (Fig. 5h, i), indicating that V3 can also suppress TGS in the host plant.

To confirm the effect of V3 on DNA methylation, the level of genome-wide methylation in transgenic V3-YFP plants was examined by digestion with a methylation-dependent restriction enzyme, *Mcr*BC[41]. As presented in Fig. 5j, genomic DNA from two independent V3-YFP lines was completely digested by the methylation-independent restriction endonuclease *Dra*I, but only partially digested by *Mcr*BC, in sharp contrast to the genomic DNA from WT plants. This indicates a lower level of DNA methylation in transgenic plants expressing V3, further supporting a function of this viral protein as a TGS suppressor.

In summary, our results demonstrate that V3 is a newly described protein encoded by TYLCV, which preponderantly localizes in the cis-Golgi, significantly contributes to virulence, and functions as a suppressor of both PTGS and TGS.

## Discussion

Geminivirus-encoded proteins have so far been identified taking into consideration an arbitrary threshold of 10 kDa, below which potential proteins were discarded. However, it is increasingly clear that small proteins and peptides play relevant roles in biological systems, hence calling for the reconsideration of this criterion. Here, we show that geminiviral genomes contain additional ORFs beyond those previously described, at least some of which are conserved within members of a given genus, hinting at functional relevance. Using TYLCV as a model, we demonstrate that at least some of these conserved additional small ORFs are expressed during the infection, and that the proteins they encode localize in specific subcellular compartments. Interestingly, in this subset of selected proteins novel localizations, not described for any of the other TYLCV-encoded proteins, are represented, including the Golgi apparatus (ORF4, 5, and 6) and mitochondria (ORF 3). Also of note, some of these proteins (ORF2, 4, and 6) harbor transmembrane domains, which are not present in any of the "canonical" proteins from TYLCV. These results indicate that the repertoire of geminiviral proteins can be expanded, and that the new additions to the viral proteomes will most likely perform additional virulence functions and/or employ alternative molecular mechanisms to those exhibited by the previously described proteins. We might still be, therefore, far from getting a comprehensive overview of the plant-geminivirus molecular interaction landscape, which will require the detailed study of potentially multiple small ORFs that have so far been neglected.

In this work, we selected the largest of the additional ORFs found in TYLCV, which we name *V3*, to be used as a proof-of-concept of the potential functionality of previously overlooked viral small proteins. Strikingly, the lack of V3 negatively impacted the viral infection in two different hosts, *N. benthamiana* and tomato, demonstrating that V3 has a biological function. However, V3 is not essential, since a V3 null mutant virus can still accumulate and establish a systemic infection.

The V3 protein is mostly localized in the cis-Golgi, a novel localization for a geminivirus-encoded protein. Remarkably, V3 can suppress both PTGS and TGS, an ability that may underlie its virulence-promoting effect on TYLCV and PVX. One intriguing question is how V3 can exert this effect from the Golgi apparatus. Interestingly, connections have been drawn between the

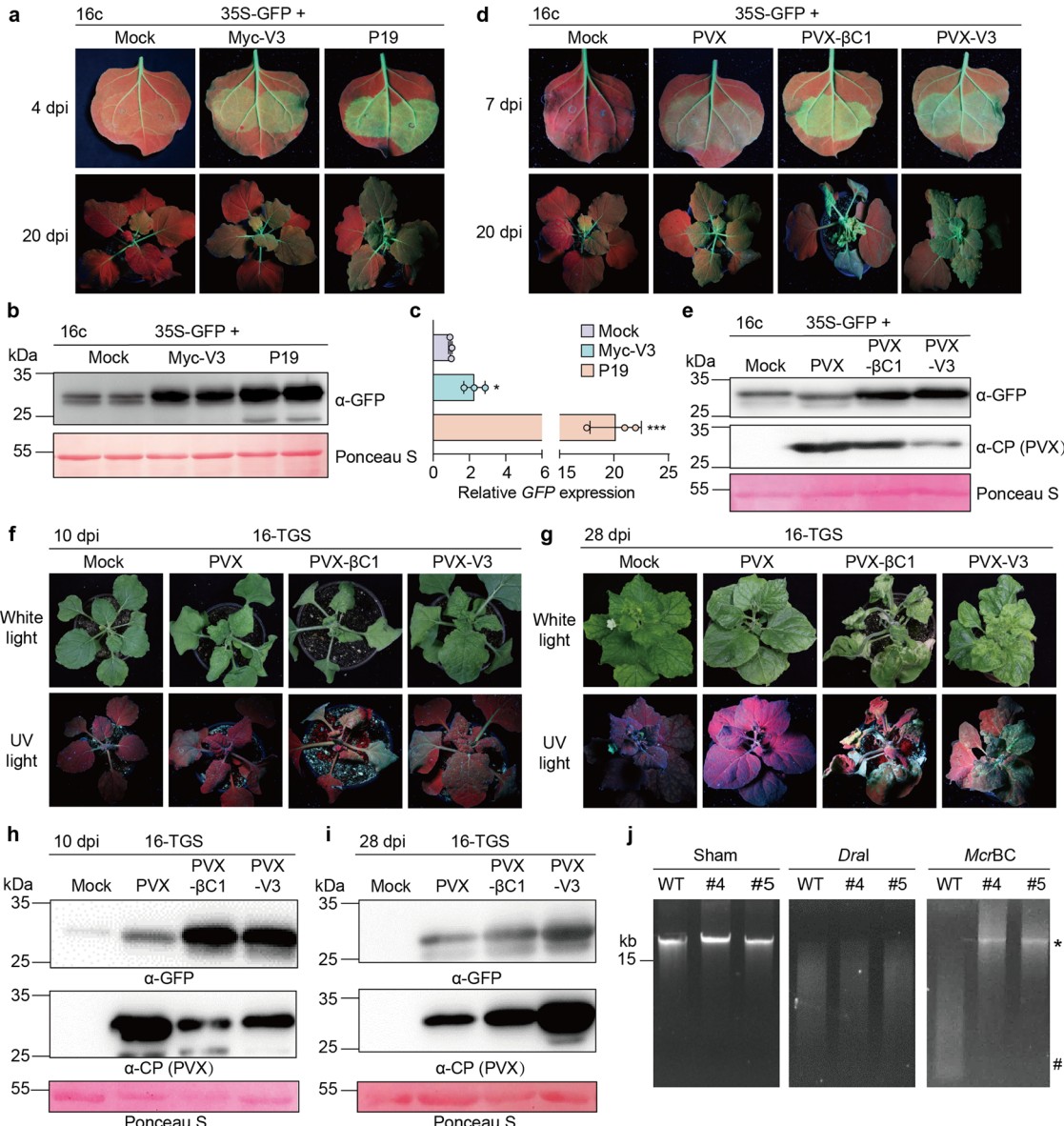

**Fig. 5 V3 functions as a suppressor of PTGS and TGS. a** Transgenic 16c *N. benthamiana* plants co-infiltrated with constructs to express GFP (35S-GFP) and Myc-V3, P19 (as positive control), or mock (empty vector, as negative control) at 4 dpi (upper panel) or 20 dpi (lower panel) under UV light. **b** Western blot showing the GFP accumulation in inoculated leaves from (**a**) at 4 dpi. The corresponding Ponceau S staining of the large RuBisCO subunit serves as a loading control. This experiment was repeated three times with similar results; representative images are shown. **c** Relative GFP mRNA accumulation in inoculated leaves from (**a**) at 4 dpi measured by qRT-PCR. Data are the mean of three independent biological replicates. Error bars represent SD. An asterisk indicates a statistically significant difference according to unpaired Student's *t* test (two-tailed), * $p < 0.5$, *** $p < 0.001$. *NbActin2* was used as the internal reference. **d** Transgenic 16c *N. benthamiana* plants co-infiltrated with constructs to express GFP (35S-GFP) and PVX, PVX-V3, PVX-βC1 (as positive control), or mock (infiltration buffer, as negative control) at 7 dpi (upper panel) or 20 dpi (lower panel) under UV light. **e** Western blot showing the GFP accumulation in inoculated leaves from (**d**) at 7 dpi. The corresponding Ponceau S staining of the large RuBisCO subunit serves as loading control. This experiment was repeated three times with similar results; representative images are shown. **f, g** Symptoms of 16-TGS *N. benthamiana* plants infected with PVX, PVX-V3, PVX-βC1 (as positive control), or mock-inoculated under white light or UV light at 10 dpi (**f**) and 28 dpi (**g**). **h, i** Western blot showing accumulation of GFP and PVX CP in systemically infected leaves from (**f**) and (**i**). The corresponding Ponceau S staining of the large RuBisCO subunit serves as loading control. This experiment was repeated three times with similar results; representative images are shown. **j** DNA methylation analysis by restriction enzyme digestion in V3-YFP transgenic *N. benthamiana* plants. Genomic DNA extracted from WT *N. benthamiana* or two independent V3-YFP transgenic lines (#4 and #5) was digested with the methylation-dependent restriction enzyme *Mcr*BC and the methylation-insensitive enzyme *Dra*I. "Sham" indicates a mock digestion with no enzyme added. The positions of undigested input genomic DNA is indicated with an asterisk; the position of the *Mcr*BC-digested products is indicated with a hashtag. This experiment was repeated three times with similar results; representative images are shown. The original data from all experiments and replicates can be found in the Source data file.

endomembrane system and RNA silencing (reviewed in Kim et al., 2014). In the model plant *Arabidopsis thaliana*, electron microscopy unveiled an enrichment of the Argonaute protein AGO1, a central player in PTGS, in close proximity to Golgi[42]; another Argonaute protein, AGO7, which has been recently shown to play a role in anti-viral defense[43], also co-purifies with membranes and concentrates in cytoplasmic bodies linked to the ER/Golgi endomembrane system[44]. It seems therefore plausible that this subcellular localization is permissive for a direct targeting of RNA silencing, although further experiments will be necessary to uncover the exact molecular mechanism underlying this activity of V3.

PTGS and TGS are arguably the main plant defense mechanisms against geminiviruses. This idea is supported by the fact that, despite limited coding capacity, a given geminivirus species can produce several proteins that target these processes. TYLCV encodes at least three proteins capable of acting as PTGS suppressors, namely C2, C4, and V2[9,16,29], and at least two, Rep and V2, capable of suppressing TGS[25–27,45]; these proteins exert their functions through non-overlapping mechanisms. Only one of the viral proteins, V2, has been described as a simultaneous suppressor of PTGS and TGS, as observed for V3. All of these silencing suppressors encoded by TYLCV are essential for the infection, although this may be due to their contribution to additional virulence activities, enabled by their multifunctional nature. Similarly, it is possible that V3 exerts additional, yet-to-be described functions during the viral infection.

Why a given geminivirus species needs multiple proteins targeting the same pathway is a thought-provoking question. Since the viral infection is a process, the temporal dimension must be considered: geminiviral genes can be classified as early or late, depending on the timing of their expression, with the strongest PTGS and TGS suppressor, V2, being a late gene, probably due to the requirement of another viral protein, C2, to activate its expression. The V3 promoter was active in the absence of the infection, which suggests that it can be expressed as an early gene. The early expression of V3 would guarantee the availability of a PTGS and TGS suppressor during the time between the synthesis of the dsDNA replicative intermediate and the expression of V2 later in the cycle, enhancing the effectiveness of viral accumulation and spread. Nevertheless, further work will be required to acquire a full understanding of the potential breadth of functions exerted by V3 and of the underpinning molecular mechanisms.

## Methods

**Plant materials**. *N. benthamiana* and *Solanum lycopersicum* (tomato) plants were grown in a growth chamber with 60% relative humidity and a 16 h:8 h light:dark, 25 °C:18 °C regime. The transgenic GFP 16c line was kindly provided by David C. Baulcombe (University of Cambridge, UK)[38]; 16-TGS plants were described previously[46]; the transgenic RFP-H2B line was kindly shared by Michael M. Goodin (University of Kentucky, USA)[47].

The *Agrobacterium tumefaciens* strain EHA105 containing the pEarleygate101:V3-YFP construct was used to generate 35S:V3-YFP transgenic *N. benthamiana* lines by leaf disc transformation as previously described[48]. In brief, the pEarleygate101:V3-YFP vector was transformed into the *A. tumefaciens* EHA105 strain and co-cultured with *N. benthamiana* leaf discs. Transformants were selected on medium containing 0.005% Basta (Coolaber, China) and then transferred to shoot regeneration medium for rooting culture. Transgenic plants were transferred to soil and screened by PCR with specific primers (Supplementary Table 1). The accumulation of the V3 protein and the corresponding mRNA were confirmed by western bot and qRT-PCR, respectively.

**Agroinfiltration and viral inoculation**. For *A. tumefaciens*-mediated transient expression in *N. benthamiana*, plasmids were transformed into the EHA105 strain by the freeze-thaw method (Fig. 3a–h lower panel, 4, 5, Supplementary Fig. 10b, c, Supplementary Fig. 12) or to the GV3101 strain through electroporation (Figs. 2, 3h upper panel, Supplementary Fig. 8, Supplementary Fig. 9, Supplementary Fig. 10a, d). *Agrobacterium* cultures were resuspended in infiltration buffer (10 mM MgCl$_2$, 10 mM MES (pH 5.6), and 100 μM acetosyringone) to an OD$_{600}$ = 0.1–0.5, then infiltrated into the adaxial side of four-week-old *N. benthamiana* leaves with a

needle-less syringe. For viral inoculation, two-week-old *N. benthamiana* plants or tomato plants at the two-leaf stage were infiltrated with *Agrobacterium* cultures carrying the TYLCV-BJ (MN432609) infectious clone. For experiments that required co-infiltration, *Agrobacterium* suspensions carrying different constructs were mixed at 1:1 ratio before infiltration.

**Plasmid construction**. Viral open reading frames (ORFs) from the TYLCV (TYLCV-Alm, Accession No. AJ489258) genome were cloned in the pENTR™/D-TOPO® vector (Thermo Scientific) (for ORF1, ORF2, ORF4, ORF5, ORF6/V3) or the pDONR™/Zeo vector (Thermo Scientific) (for ORF3) without a stop codon. The binary plasmids to express GFP-fused viral proteins were generated by sub-cloning (Gateway LR reaction, Thermo Scientific) the viral ORFs from the corresponding entry vectors into pGWB505[49]. To generate the constructs to express V3-YFP, YFP-V3, or Myc-V3, the full-length V3 ORF was obtained from TYLCV (TYLCV-BJ, Accession No. MN432609) and recombined into the binary destination vectors pEarleygate101, pEarleygate104, or pEarleygate203, respectively[50]. Please note that the V3/ORF6 protein sequences from TYLCV-Alm and TYLCV-BJ are identical.

To generate the construct to express SYP32-RFP (as a cis-Golgi marker), the gene encoding SYNTAXIN OF PLANTS 32 (SYP32) was amplified from *A. thaliana* cDNA, cloned into the pENTR™/D-TOPO® vector (Thermo Scientific), and sub-cloned into pGWB554[49] using a Gateway LR reaction (Thermo Scientific). The construct to express RFP-HDEL is from[51], and the ones to express mCherry-HDEL and Man49-mCherry are from[52].

To generate the constructs used for the analysis of promoter activity, the 500-nt sequence upstream of the ORF1, ORF2, ORF4, ORF5, ORF6/V3, and C4 ATG or the 570-nt sequence upstream of the ORF3 ATG, were PCR-amplified and cloned into the pDONR™/Zeo vector (Thermo Scientific). These sequences were then sub-cloned into pGWB504[49] by a Gateway LR reaction to generate pORF-GFP. In addition, the 833-nt sequence upstream of the V3/ORF6 ATG was PCR-amplified and cloned with pINT121-GUS digested with *Hind*III and *Bam*HI to generate pINT121-V3-GUS (pV3-GUS), or into pCHF3-GFP digested with *Eco*RI and *Sac*I to generate pCHF3-V3-GFP (pV3-GFP) using In-Fusion Cloning according to the manufacturer's instructions. The full-length V3 ORF was inserted into the PVX vector digested with *Cla*I and *Sal*I to generate PVX-V3. The pCHF3-35S-GFP, pCHF3-p19, PVX-βC1, and a PVX-based expression vector for PTGS suppression assays have been described previously[53,54], as has the PVX-based expression vector PVX-βC1 for TGS suppression assays[55]. All primers used in this study can be found in Supplementary Table 1.

**Sequence analysis**. The ViralORFfinder platform (see Code availability) was constructed by Shiny, an R package used to build interactive web applications (https://shiny.rstudio.com). The NCBI ORFfinder (https://www.ncbi.nlm.nih.gov/orffinder/) was used to identify ORFs for each uploaded virus, and the R package Gviz[56] was used for visualization; for Supplementary Figs. 1 and 3, 1.2-mer genomic sequences were used. To investigate the conservation of an ORF-encoded protein of choice, this tool creates a small database with the translated sequences from the inputted viral sequences, and BLASTp is used to identify proteins with high identity (e-value ≤ 0.05). Phylogenetic trees were obtained by the R package DECIPHER and visualized by ggtree. Multiple sequence alignments were constructed by ClustalW and visualized by the R package ggmsa (https://cran.r-project.org/web/packages/ggmsa/vignettes/ggmsa.html). Names and NCBI accession numbers of virus species used in this work are listed in Supplementary Tables 2–6.

**Prediction of domains or signals in protein sequences**. The prediction of transmembrane domains (TM) was performed by TMHMM (http://www.cbs.dtu.dk/services/TMHMM/) and Phobius (https://phobius.sbc.su.se/). The prediction of nuclear localization signal (NSL) was performed by cNLS Mapper (http://nls-mapper.iab.keio.ac.jp/cgi-bin/NLS_Mapper_form.cgi[57,58]), The prediction of chloroplast transit peptide (cTP) was performed by ChloroP (http://www.cbs.dtu.dk/services/ChloroP/).

**Confocal microscopy**. Confocal microscopy was performed using a Leica TCS SP8 point scanning confocal microscope (Figs. 2, 3h upper panel, Supplementary Figs. 8, 9, 10a, and 10d) or Zeiss LSM980 confocal microscope (Carl Zeiss) (Figs. 3e, 3h lower panel, Supplementary Fig. 10b, c), with the preset settings for GFP (Ex: 488 nm, Em: 500-550 nm), RFP (Ex: 561 nm, Em: 570-620 nm), YFP (Ex :514 nm, Em: 515–570 nm), or mCherry (Ex: 594 nm, Em: 597–640 nm). For co-localization imaging, the sequential scanning mode was used.

**Mitochondrial staining**. To visualize mitochondria, staining with 250 nM Mito-Tracker® Red CMXRos (Invitrogen) was used. The chemical was infiltrated 10-30 min before imaging. The stock solution (1 mM) was prepared by dissolving the corresponding amount of MitoTracker® in dimethylsulfoxide (DMSO). The working solution was prepared by diluting the stock solution in water or infiltration buffer (10 mM MgCl$_2$, 10 mM MES (pH 5.6), and 100 μM acetosyringone). The MitoTracker® red fluorescence was imaged using a Leica TCS SP8 point scanning confocal microscope with the following settings: Ex: 561 nm, Em: 570-620 nm.

**DNA and RNA extraction and qPCR/qRT-PCR.** Total DNA was extracted from infected plants using the CTAB method. Total RNA was extracted from collected plant leaves using TaKaRa MiniBEST Universal RNA Extraction Kit (Takara, Japan). 1 ug of total RNA was reverse-transcribed into cDNA using PrimeScript™ RT reagent Kit with gDNA Eraser (Perfect Real Time) (Takara, Japan). qPCR or qRT-PCR was performed using TB Green® Premix Ex Taq™ II (Takara, Japan). 25S RNA and NbActin2 were used as internal references for DNA and RNA normalization, respectively.

**5′ rapid amplification of cDNA ends (RACE).** Total RNA extracted from TYLCV (TYLCV-BJ, Accession No. MN432609)-infected N. benthamiana plants was used for 5′ RACE with SMARTer RACE 5′/3′ Kit (Takara, Japan) according to the manual booklet.

**3,3′-diaminobenzidine (DAB) staining.** For DAB staining, systemic leaves of infected plants were incubated in DAB solution (1 mg/mL, pH 3.8) for 10 h at 25 °C, then boiled for 5-10 min and decolorized in 95% ethanol.

**Protein extraction and western blotting.** Total protein was isolated from infiltrated leaf patches with protein extraction buffer (containing 50 mM Tris-HCl (pH 6.8), 4.5% (m/v) SDS, 7.5% (v/v) 2-Mercaptoethanol, 9 M carbamide). Immunoblotting was performed with primary mouse polyclonal antibodies, followed by anti-mouse IgG HRP-linked antibodies (1:5000; Cell Signaling Technology, Cat#: 7076, USA); primary antibodies used are as follows: anti-GFP (1:5000; ROCHE, Cat#: 11814460001, USA), and custom-made anti-PVX CP (1:5000) and anti-TYLCV CP (1:5000)[59,60].

## Data availability

All data generated or analyzed during this study are included in this article (and its supplementary information files). Source data are provided with this paper.

## Code availability

The code for developing ViralORFfinder can be found on GitHub at https://github.com/mayupsc/vORFfinder2.0 (https://doi.org/10.5281/zenodo.4950738)[61]

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

## Acknowledgements

This work was supported by the National Natural Science Foundation of China (31930089 and 31720103914), the Strategic Priority Research Program of the Chinese Academy of Sciences (Grant No. XDB27040206), and the Shanghai Center for Plant Stress Biology from the Chinese Academy of Sciences. The authors thank all members of Rosa Lozano-Duran's lab and Alberto Macho's lab for fruitful discussions, Xinyu Jian, Aurora Luque, and the PSC Cell Biology Facility for technical assistance, Alberto Macho for critical reading of the manuscript, and Dr. Michael M. Goodin (University of Kentucky, USA) and Prof. David Baulcombe (University of Cambridge) for kindly sharing materials.

## Author contributions

Conceptualization: R.L.-D., F.L., X.Z.; Investigation: P.G., H.T., S.Z., H.L., H.L., Y.M., X.Z., J.R., X.F.; Writing: P.G., H.T., R.L.-D., F.L., X.Z.; Visualization: P.G., H.T.; Supervision: R.L.-D., F.L., X.Z.; Funding acquisition: R.L.-D., F.L., X.Z.

## Competing interests

The authors declare no competing interests.
