## [Peer Review File · Nature Communications]

REVIEWER COMMENTS

Reviewer #1 (Remarks to the Author):

Manuscript NCOMMS-21-08248 describes the identification of hitherto undescribed open reading frames (ORFs) in the genome of geminiviruses. Select proteins encoded by some of these small ORFs have a specific cellular localization such as the Golgi apparatus or the mitochondria; and some of them carry transmembrane domains. One of these ORFs in the genome of tomato yellow leaf curl virus (TYLCV) is named V3. Protein V3 is essential to the infectious cycle in planta by localizing to the Golgi apparatus and acting as a suppressor of RNA silencing at the transcriptional and post-transcriptional levels. This research is novel, thoroughly conducted and of high interest not only to the plant virus community but also to the plant biology community at large. It may inspire others to reconsider the coding capacity of often-ignored small ORFs. The methodologies are appropriate and the conclusions are sound. In addition, the manuscript is very well written.

Specific comments:

Line 1: Change additional to small

Line 20: ... proteins are traditionally identified by applying ...

Line 24: Change novel to small

Line 33: Change life to infectious

Line 61: What are things? Could a scientifically-sound term be used?

Line 68: A reference could be added here

Line 69: ... monopartite begomovirus and the causal agent of ...

Line 76: What do PCNA and NAC stand for?

Lines 81-82: ... suppressor of PTGS, as well as transcriptional gene silencing (TGS), and mediates the nuclear export of the CP ...

Line 91: ... during viral infection ...

Line 98: Change intersection to interaction

Line 126: ... selected the next six ORFs based on size and prevalence ...

Line 141: ... leaves. Confocal microscopy ...

Line 145: What does HDEL stand for?

Lines 237-238: Tomato yellow leaf curl China betasatellite (in italics) should read tomato yellow leaf curl China betasatellite (no italics)

Line 301: Change novel to small

Line 365: RFP=H2B or RFP-HDEL as indicated on line 145?

Line 380: It would be nice to cite a reference for the infectious clone

Supplementary Figure 1, caption: Change Bean golden yellow mosaic virus (in italics) to bean golden mosaic virus (no italics)

Supplementary Figure 1, caption: Change Genbank to GenBank

Supplementary Figure 2, caption: Change Beet curly top virus (in italics) to beet curly top virus (no italics)

Supplementary Figure 3, caption: Change Maize streak virus (in italics) to maize streak virus (no italics)

Supplementary Figure 34 caption: Change Tomato yellow leaf curl virus (in italics) to tomato yellow leaf curl virus (no italics)

Supplementary Tables 2-6: Virus names should not be italicized

Reviewer #2 (Remarks to the Author):

SUMMARY

Manuscript 300619 reports on the identification and biological characterization of a new protein V3 from Tomato yellow leaf curl virus (TYLCV). The research is based on a previous publication that predicted that geminiviral genomes encode additional proteins. In this manuscript, the authors used a computational approach to predict additional proteins in TYLCV. Presence of those proteins was confirmed *in vivo*, and V3 was characterized: it is a previously unknown gene silencing suppressor that is necessary for TYLCV infection. Results are informative in the study of plant-virus

interactions. Minor corrections are suggested.

MAIN CONTRIBUTIONS

1. A computational approach was used to predict additional proteins in TYLCV.
2. Presence of those proteins was confirmed *in vivo*.
3. V3 is a previously unknown gene silencing suppressor that is necessary for TYLCV infection.

POINTS THAT NEED TO BE CLARIFIED

1. Promoters driving expression of the additional proteins. Lanes 154 to 163 present an experiment to test promoter activity of TYLCV genomic sequences upstream of the start codon for new open reading frames 1 through 6. None of the constructs, including V3, could drive transcription in the absence of the virus. Some could drive transcription in the presence of the virus.

In contrast, in lanes 185 through 193 the authors report that genomic sequences upstream of V3 provide promoter activity.

An explanation is needed for this contrasting results.

2. The title indicates that the manuscript is about geminiviruses encoding additional proteins with virulence function. Figure 1 shows protein prediction. This result is redundant with a recent publication from the same group. The rest of the paper is focused on V3, its detection and biological role.

The current title does not accurately reflect the new findings.

Response to reviewers

We would like to thank the reviewers for their positive assessment of our manuscript. Following their advice, we have made a number of changes to our manuscript, as highlighted in the new version of the text; a point-by-point response to the reviewers' comments is provided below.

Reviewer #1 (Remarks to the Author):

Manuscript NCOMMS-21-08248 describes the identification of hitherto undescribed open reading frames (ORFs) in the genome of geminiviruses. Select proteins encoded by some of these small ORFs have a specific cellular localization such as the Golgi apparatus or the mitochondria; and some of them carry transmembrane domains. One of these ORFs in the genome of tomato yellow leaf curl virus (TYLCV) is named V3. Protein V3 is essential to the infectious cycle in planta by localizing to the Golgi apparatus and acting as a suppressor of RNA silencing at the transcriptional and post-transcriptional levels. This research is novel, thoroughly conducted and of high interest not only to the plant virus community but also to the plant biology community at large. It may inspire others to reconsider the coding capacity of often-ignored small ORFs. The methodologies are appropriate and the conclusions are sound. In addition, the manuscript is very well written.

Specific comments:

Line 1: Change additional to small

- Following the reviewer's advice, we have added "small" to the title. We have maintained the adjective "additional", since we believe making this point is important – since all proteins encoded by geminiviruses, including the ones previously described, may be considered small, and also present specific subcellular localization and virulence function, if the fact that additional ones are described in this work is not specifically mentioned the title would not convey the desired message.

Line 20: ... proteins are traditionally identified by applying ...

- We have changed this sentence as suggested by the reviewer – thanks.

Line 24: Change novel to small

- Changed – thanks.

Line 33: Change life to infectious

- Changed – thanks.

Line 61: What are things? Could a scientifically-sound term be used?

- Here, we were using "among other things" as an expression. We have now replaced "things" with "processes".

Line 68: A reference could be added here

- We think it is not necessary for a reference in this place.

Line 69: ... monopartite begomovirus and the causal agent of ...

- Changed – thanks.

Line 76: What do PCNA and NAC stand for?

- We have now indicated the full name for PCNA (PROLIFERATING CELL NUCLEAR ANTIGEN), and specifically mentioned SINAC1 as the transcription factor from the NAC family interacting with C3 (Selth et al., 2005).

Lines 81-82: ... suppressor of PTGS, as well as transcriptional gene silencing (TGS), and mediates the nuclear export of the CP ...

- We have now added the missing “as” – thanks.

Line 91: ... during viral infection ...

- Since “during the viral infection” seems to be a more commonly used formula, we have refrained from making this change.

Line 98: Change intersection to interaction

- We have now rephrased this sentence to read “...Taken together, our results indicate that geminiviruses encode additional proteins to the ones described to date, which may largely expand the geminiviral proteome and hence the intersection of these viruses with the host cell.”

Line 126: ... selected the next six ORFs based on size and prevalence ...

- Changed – thanks.

Line 141: ... leaves. Confocal microscopy ...

- Changed.

Line 145: What does HDEL stand for?

- HDEL is a C-terminal ER retention protein signal [Histidine (H) - Aspartic acid (D) - Glutamic acid (E)- Leucine (L)].

Lines 237-238: Tomato yellow leaf curl China betasatellite (in italics) should read tomato yellow leaf curl China betasatellite (no italics)

- Changed.

Line 301: Change novel to small

- We have replaced “novel” with “additional small” (see previous comment).

Line 365: RFP=H2B or RFP-HDEL as indicated on line 145?

- In this work, we use both a construct to express RFP-HDEL (as ER marker) and transgenic *Nicotiana benthamiana* plants expressing RFP-H2B (as nuclear marker).

Line 380: It would be nice to cite a reference for the infectious clone

- The TYLCV-BJ infectious clone is not yet published; nevertheless, the manuscript in which it is described is already in press (Gong, P., Zhao, S., Zhou, X., & Li F. *The establishment of TYLCV-BJ isolate infectious clone*. J Plant Protec (In Chinese).

Supplementary Figure 1, caption: Change Bean golden yellow mosaic virus (in italics) to bean golden mosaic virus (no italics)

- Changed.

Supplementary Figure 1, caption: Change Genbank to GenBank

- Changed.

Supplementary Figure 2, caption: Change Beet curly top virus (in italics) to beet curly top virus (no italics)

- Changed.

Supplementary Figure 3, caption: Change Maize streak virus (in italics) to maize streak virus (no italics)

- Changed.

Supplementary Figure 34 caption: Change Tomato yellow leaf curl virus (in italics) to tomato yellow leaf curl virus (no italics)

- Changed.

Supplementary Tables 2-6: Virus names should not be italicized

- Changed.

Reviewer #2 (Remarks to the Author):

SUMMARY

Manuscript 300619 reports on the identification and biological characterization of a new protein V3 from Tomato yellow leaf curl virus (TYLCV). The research is based on a previous publication that predicted that geminiviral genomes encode additional proteins. In this manuscript, the authors used a computational approach to predict additional proteins in TYLCV. Presence of those proteins was confirmed *in vivo*, and V3 was characterized: it is a previously unknown gene silencing suppressor that is necessary for TYLCV infection. Results are informative in the study of plant-virus interactions. Minor corrections are suggested.

MAIN CONTRIBUTIONS

1. A computational approach was used to predict additional proteins in TYLCV.
2. Presence of those proteins was confirmed *in vivo*.
3. V3 is a previously unknown gene silencing suppressor that is necessary for TYLCV infection.

POINTS THAT NEED TO BE CLARIFIED

1. Promoters driving expression of the additional proteins. Lanes 154 to 163 present an experiment to test promoter activity of TYLCV genomic sequences upstream of the start codon for new open reading frames 1 through 6. None of the constructs, including V3, could drive transcription in the absence of the virus. Some could drive transcription in the presence of the virus.

In contrast, in lanes 185 through 193 the authors report that genomic sequences upstream of V3 provide promoter activity.

An explanation is needed for this contrasting results.

- We thank the reviewer for pointing out this apparent discrepancy. In the initial promoter assay, the 500 bp-fragment upstream of the V3 ATG was used, and no promoter activity was detected in the absence of the virus; the length of the tested sequences is now specified in line 157, for clarity. Later, a longer, 833-bp fragment was tested, and this proved sufficient to drive gene expression even in the absence of the virus. This is explained in lines 189-191.

2. The title indicates that the manuscript is about geminiviruses encoding additional proteins with virulence function. Figure 1 shows protein prediction. This result is redundant with a recent publication from the same group. The rest of the paper is focused on V3, its detection and biological role.

The current title does not accurately reflect the new findings.

- Our apologies, but we do not understand what the reviewer is referring to with “This result is redundant with a recent publication from the same group” – the results

contained in this manuscript have not been published elsewhere, with the exception of the pre-print version of the manuscript available on bioRxiv (<https://www.biorxiv.org/content/10.1101/2021.03.01.433473v1>). We think our title 'Geminiviruses encode additional small proteins with specific subcellular localizations and virulence function' include our findings.